# Positive Unlabelled Learning for Satellite Images'Time Series Analysis: An Application to Cereal and Forest Mapping

**Johann Desloires** [1,2,*]**, Dino Ienco** [1] , **Antoine Botrel** [2] **and Nicolas Ranc** [2]

1 INRAE, UMR TETIS, University of Montpellier, 34000 Montpellier, France; dino.ienco@inrae.fr
2 Syngenta Seeds, 31790 Saint-Sauveur, France; antoine.botrel@syngenta.com (A.B.);
  nicolas.ranc@syngenta.com (N.R.)
* Correspondence: johann.desloires@syngenta.com

**Abstract:** Applications in which researchers aim to extract a single land type from remotely sensed data are quite common in practical scenarios: extract the urban footprint to make connections with socio-economic factors; map the forest extent to subsequently retrieve biophysical variables and detect a particular crop type to successively calibrate and deploy yield prediction models. In this scenario, the (positive) targeted class is well defined, while the negative class is difficult to describe. This one-class classification setting is also referred to as positive unlabelled learning (PUL) in the general field of machine learning. To deal with this challenging setting, when satellite image time series data are available, we propose a new framework named positive and unlabelled learning of satellite image time series (PUL-SITS). PUL-SITS involves two different stages: In the first one, a recurrent neural network autoencoder is trained to reconstruct only positive samples with the aim to higight reliable negative ones. In the second stage, both labelled and unlabelled samples are exploited in a semi-supervised manner to build the final binary classification model. To assess the quality of our approach, experiments were carried out on a real-world benchmark, namely *Haute-Garonne*, located in the southwest area of France. From this study site, we considered two different scenarios: a first one in which the process has the objective to map *Cereals/Oilseeds* cover versus the rest of the land cover classes and a second one in which the class of interest is the *Forest* land cover. The evaluation was carried out by comparing the proposed approach with recent competitors to deal with the considered positive and unlabelled learning scenarios.

**Keywords:** land cover mapping; positive unlabelled learning; satellite image time series; deep learning

## 1. Introduction

Currently, modern remote sensing systems provide image acquisitions describing the Earth's surface at a high spatial resolution and revisit time period. A notable example is the Copernicus programme, which, through the Sentinel-2 mission, provides optical multispectral images within the visible and near-infrared regions (electromagnetic spectrum) with a spatial resolution between 10 m and 60 m and a revisit time period of approximately 5 days [1]. Such a stream of information can be profitably organised as satellite image time series (SITS) and can support a wide range of application domains, such as ecology [2], agriculture [3], mobility, health, risk assessment [4], land management planning [5], and forest [6], and natural habitat monitoring [7]. Due to the large panel of applications that can be supported by such data, it constitutes a valuable source of information to monitor the dynamics of the Earth's surface.

One of the main tasks related to SITS data analysis is associated with land cover mapping, where a classification model is learned to make the connection between satellite data (i.e., SITS) and the associated land cover classes [5]. SITS data capture the temporal dynamics exhibited by land cover classes, thus resulting in a more effective discrimination among them [8].

Land cover maps can be the final outcome of the analysis or it can constitutes an intermediate step of a more general pipeline [9]. Examples of the latter case are the mapping of a specific culture versus the rest of the land cover classes to successively calibrate and deploy a yield prediction model [9], distinguish between forest areas and the rest of the land cover classes to subsequently retrieve biophysical variables such as canopy height or biomass volume [10], or extract the settlement footprint detecting urban vs. non-urban land covers to make a connection with socio-economic factors [11]. In these particular applications, the objective is to obtain binary land cover maps where one category (the positive one) is well defined since it is associated with the land cover of interest, while the other category (the negative one) is difficult to describe since it gathers together the rest of the possible land cover classes present in a particular study area. This one-class classification setting is also referred to as positive unlabelled learning (PUL) [12] in the general field of machine learning. Different from the standard supervised classification context, in the positive unlabelled learning setting, training data are composed of a set of positive samples and a set of unlabelled ones with the latter involving both positive and negative samples. The objective of a PUL framework is to build a binary classification model that can make predictions on unseen data distinguishing between positive and negative samples.

With the aim to focus on the analysis of SITS data, in this work, we propose a new framework, named PUL-SITS, to deal with positive and unlabelled learning for land cover mapping from satellite image time series. PUL-SITS involves two different stages: In the first one, a recurrent neural network autoencoder is trained only on positive samples. Successively, the same autoencoder model is employed to filter out reliable negative samples from the unlabelled ones. In the second stage, both labelled (positive and reliable negative) and unlabelled samples are exploited in a semi-supervised manner to build the final binary classification model. In this stage, the unlabelled samples are employed to force consistency between the prediction of the classification model and a variant of the same network trained on the SITS data reconstructed by the recurrent neural network autoencoder model.

To assess the effectiveness of PUL-SITS, we considered two different scenarios involving data coming from a particular study area (*Haute-Garonne*) located in the southwest area of France. More precisely, the *Haute-Garonne* study area is heavily characterized by the *Cereals/Oilseeds* and *Forest* land cover classes. For this reason, we designed a first scenario in which the *Cereals/Oilseeds* land cover class has the role of the positive class, while the rest of the land cover classes play the role of the negative class. In the second scenario, the *Forest* land cover class is targeted as the positive class. Experiments were carried out comparing the proposed approach w.r.t. well-suited competitors to deal with the considered positive and unlabelled learning scenarios. The remainder of this paper is structured as follows: First, Section 2 reviews related work in literature at the intersection of positive and unlabelled learning and remote sensing. The proposed framework is described in Section 3. The study site and the associated data are presented in Section 4. The experimental settings, as well as the evaluation are reported and discussed in Section 5. Finally, Section 6 draws the conclusion.

## 2. Related Work

Regarding the remote sensing domain, Reference [13] investigated the use of positive and unlabelled learning to cope with urban footprint mapping considering high-spatial-resolution aerial photographs. The PUL algorithm trains a classifier on positive and unlabelled data, estimating the probability that a positive training sample has been labelled, and finally, it generates binary predictions for test samples using an adjusted threshold. Reference [14] proposed an ensemble approach to deal with one-class classification under a PUL setting. Here, the class of interest was still the urban land cover, and the approach was evaluated considering different sensors (Landsat 8, WorldView-3, and Gaofen-1). Similar to [13], also in this case, the classification method was deployed on single-date images. In [15], the authors proposed to employ positive and unlabelled learning to characterize landslide susceptibility from several features extracted from multi-source data (elevation,

slope angle, curvature, slope aspect, normalized difference vegetation index, soil type, loess erosion intensity, and topographic wetness index). Recently, Reference [16] evaluated PUL-based classifiers on different land cover classes (i.e., trees, bare soil), highlighting that the proposed strategy is well adapted for a plethora of different application contexts.

Unfortunately, despite the fact that the positive unlabelled learning setting naturally fits a wide range of remote sensing applications in which researchers aim to extract a binary land cover map with a focus on a particular class of interest, all the proposed strategies were developed considering mono-date images [13,14,16], while, today, SITS data are ubiquitous in the remote sensing community. According to our literature survey, no research study has yet proposed dealing with SITS data, explicitly with the temporal dynamics carried out by such a rich source of information, under the positive and unlabelled learning setting.

## 3. Method

In this section, we briefly introduce the positive and unlabelled learning setting, and successively, we describe PUL-SITS, a new framework to deal with land cover mapping from satellite image time series under the PUL setting.

We remind the reader that we considered satellite image time series at the pixel level as the unit of analysis. We refer to a generic SITS (at the pixel level) as $TS = (x_1, \ldots, x_t, \ldots, x_T)$ where $T$ is the length of the time series and a generic element $x_t$ is a multi-variate (or multidimensional) vector with the subscript $t$, the corresponding timestamp. The PU learning setting considers a scenario in which we dispose of $D = P \cup U$ a set of samples $\{TS_i\}_{i=1}^{|D|}$ composed by a set $P$ of positive samples and a set $U$ of unlabelled ones. The set $U$ contains both positive and negative samples, but their label information is not accessible. The task of learning from both positive and unlabelled data consists of exploiting both labelled $P$ and unlabelled $U$ samples to learn a binary classification model, allowing the assignment of a binary label (positive or negative) to new, previously unseen, satellite image time series.

Regarding PUL-SITS, our framework is constituted of two different stages. The first stage is devoted to selecting a set of reliable negative samples that highly differ from the positive set. This step is addressed via a variational recurrent neural network autoencoder (VAERNN), which is used to model the set of positive samples.

In the second stage, PUL-SITS learns a satellite image time series classifier, based on the recurrent neural network (RNN) model proposed in [17], to cope with the underlying binary classification task. In this stage, a semi-supervised training procedure is designed in order to take advantage of both labelled (positive and reliable negative) and unlabelled samples when the binary classification model is trained.

### 3.1. Selection of Reliable Negative Samples

The first stage of our framework addresses the identification of a set of reliable negative ($RN$) samples. A sketch of this procedure is depicted in Figure 1. To this end, we first modelled the positive set of satellite image time series (at the pixel level) via a variational RNN autoencoder (VAERNN), then we used the trained neural network to select reliable negative $RN$ samples from the set of unlabelled satellite image time series data. To this end, the reconstruction error was used as the evaluation metric. Samples with a small reconstruction error are likely to belong to the positive class, while samples on which the reconstruction error is high are probably those representing the negative class. The use of recurrent neural networks (RNNs) to model the data distribution naturally arises due to the fact that such a model is especially tailored to managing multi-variate sequential data [18] and, in particular, compressing sequential information to extract useful multi-variate time series representations [19,20].

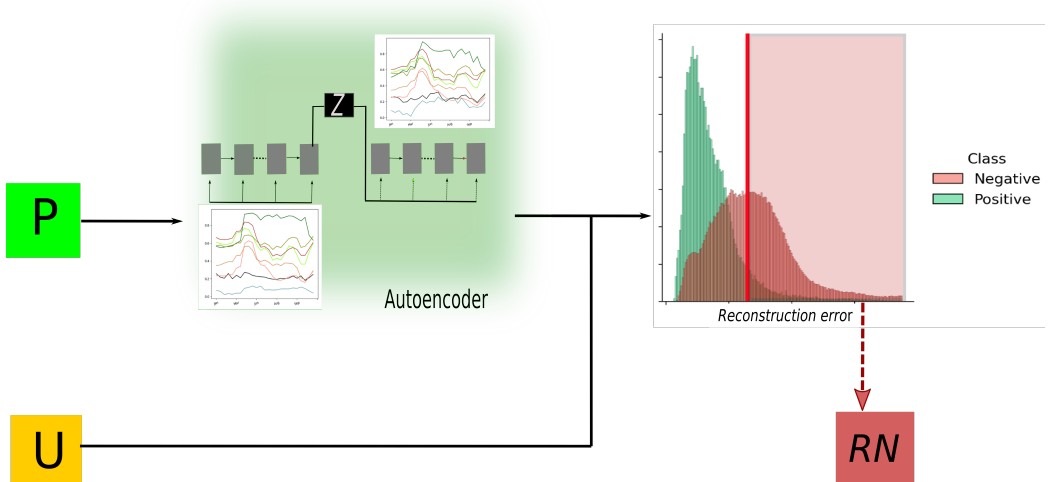

**Figure 1.** The VAERNN is trained on the set of positively labelled samples *P*. Then, the model is applied on the set of unlabelled samples *U*. Successively, the reconstruction error for each *TS* ∈ *U* is derived, and the set of reliable negative samples *RN* is identified based on the reconstruction error. The vertical red line indicates the average reconstruction error ($\mu_{rec}$) on the set of unlabelled samples *U*. Samples with a reconstruction error bigger than $\mu_{rec}$ are possible candidates for the reliable negative set *RN*.

In our case, we used a variational autoencoder [21] based on a two-level gated recurrent unit [22]. Gated recurrent units are preferred to long-short term memory [23] since the former model has demonstrated better performance than the latter one in several remote-sensing-related applications [24,25]. The gated recurrent units have fewer parameters than the LSTM cells, and this makes the learned cell more robust and easy to train; finally, GRU cells have also exhibited good performance for reconstruction tasks and, more generally, for unsupervised analysis [20,21].

The gated recurrent unit (GRU) is formally defined as follows:

$$z_t = \sigma(W_{zx}x_t + W_{zh}h_{t-1} + b_z) \tag{1}$$

$$r_t = \sigma(W_{rx}x_t + W_{rh}h_{t-1} + b_r) \tag{2}$$

$$h_t = z_t \odot h_{t-1} + \tag{3}$$

$$(1 - z_t) \odot \tanh(W_{hx}x_t + W_{hr}(r_t \odot h_{t-1}) + b_h)$$

The $\odot$ symbol indicates an elementwise multiplication, while $\sigma$ and tanh represent the sigmoid and hyperbolic tangent functions, respectively. $x_t$ is the timestamp input vector, and $h_{t-1}$ is the hidden state of the recurrent unit at time $t-1$. The different weight matrices $W_{**}$ and bias vectors $b_*$ are parameters learned during the training of the model.

This unit follows the general philosophy of modern recurrent neural network models implementing gates and cell states. The *GRU* cell has two gates, update ($z_t$) and reset ($r_t$), and one cell state, the hidden state ($h_t$). Moreover, the two gates combine the current input ($x_t$) with the information coming from the previous timestamps ($h_{t-1}$). The update gate effectively controls the tradeoff between how much information from the previous hidden state will carry over to the current hidden state and how much information of the current timestamp needs to be kept. On the other hand, the reset gate monitors how much information of the previous timestamps needs to be integrated with the current information. As each hidden cell has separate reset and update gates, they are able to capture dependencies over different time scales. Cells more prone to capturing short-term dependencies will tend to have a frequently activated reset gate, but those that capture longer-term dependencies will have update gates that remain mostly active [22]. Figure 2 visually depicts the GRU cell.

The encoder involves two GRU cells, while the decoder is symmetric to the encoder. Variational autoencoders, as opposed to standard ones, model the latent space to match a Gaussian distribution [26] (mean and variance). Such a way of compressing the input information can be more appropriate to manage complex data distributions such as in the case of satellite image time series, as recently highlighted in [27].

Furthermore, for an element at timestamps $t$ output by the autoencoder, we performed an additional linear transformation to obtain the reconstructed element $\hat{x}_t$, since, considering regression tasks, the softmax layer is commonly replaced by a fully connected layer with no activation (linear transformation) [28]. Finally, the variational autoencoder model was trained end-to-end by means of a reconstruction loss function on the set of positively labelled satellite image time series $P$:

$$LOSS_{VAERNN} = \frac{\sum_{TS \in P} HLoss(TS, Linear(VAERNN(TS)))}{|P|} \tag{4}$$

where $(Linear(VAERNN(TS)))$ is the variational recurrent neural network autoencoder followed by a linear transformation and HLoss is the Huber loss [28], defined as follows:

$$HLoss(y, \hat{y}) = \begin{cases} \frac{1}{2}(y - \hat{y})^2 & if\ |y - \hat{y}| < \delta \\ \delta|y - \hat{y}| - \frac{1}{2}\delta^2 & otherwise \end{cases} \tag{5}$$

Basically, the Huber loss behaves as the mean squared error (MSE) loss when the error is lower than a threshold $\delta$, while it mimics the mean absolute error (MAE) loss otherwise. In addition, it is also differentiable at 0. The Huber loss combines the good properties from both the MAE and MSE, and it avoids their limitations. As opposed to the MAE, it bypasses large gradient back propagation when the estimated quantity becomes closer to the real value, and different from the MSE loss, it is more robust to outliers. To manage the balance between the two components of the Huber loss function, we relied on the default parameter setting [29] in which the $\sigma$ value is set to 1.

Once the VAERNN is trained on the set of positive SITS data, it is successively deployed on the set of unlabelled time series $U$. For each time series $TS \in U$, a reconstruction error is computed via the Huber loss by comparing the original multi-variate time series with the reconstructed one.

Then, we computed the average reconstruction error $\mu_{rec}$ (the mean reconstruction error value on the set of unlabelled time series). After that, we considered multi-variate time series (belonging to the unlabelled set) with a reconstruction error greater than $\mu_{rec}$, and we named such a set $U_{\mu_{rec}}$. Finally, we randomly selected elements from $U_{\mu_{rec}}$ to build the set of reliable negative samples $RN$ with a size equal to the size of the positive set $P$ such that $|P| == |RN|$.

### 3.2. Semi-Supervised Training Procedure

Once both $P$ and $RN$ sets were available, we trained a classification model to deal with the underlying binary classification task. As a classification backbone for our PUL framework, we considered the RNN architecture recently proposed in [17] and referred to as the fully connected gated recurrent unit (FCGRU). We chose such a model due to the interesting performance it has exhibited w.r.t. other state-of-the-art approaches for land cover mapping from SITS data. The FCGRU cell extends the standard GRU cell with two additional fully connected layers that are applied to the input multi-variate time series,

with the aim to enrich the initial representation before deploying the GRU cell, as illustrated in Figure 2. More formally, the FCGRU cell is defined as follows:

$$x_{t'} = \tanh(W_2 \tanh(W_1 x_t + b_1) + b_2) \tag{6}$$

$$z_t = \sigma(W_{zx} x_{t'} + W_{zh} h_{t-1} + b_z) \tag{7}$$

$$r_t = \sigma(W_{rx} x_{t'} + W_{rh} h_{t-1} + b_r) \tag{8}$$

$$h_t = z_t \odot h_{t-1} + \tag{9}$$

$$(1 - z_t) \odot \tanh(W_{hx} x_{t'} + W_{hr}(r_t \odot h_{t-1}) + b_h)$$

where the $\odot$ symbol indicates an elementwise multiplication, while $\sigma$ and tanh represent the sigmoid and hyperbolic tangent function, respectively. $x_t$ is the timestamp input vector, and $h_{t-1}$ is the hidden state of the recurrent unit at time $t-1$. The different weight matrices $W_{**}$ and bias vectors $b_*$ are the parameters learned during the training of the model.

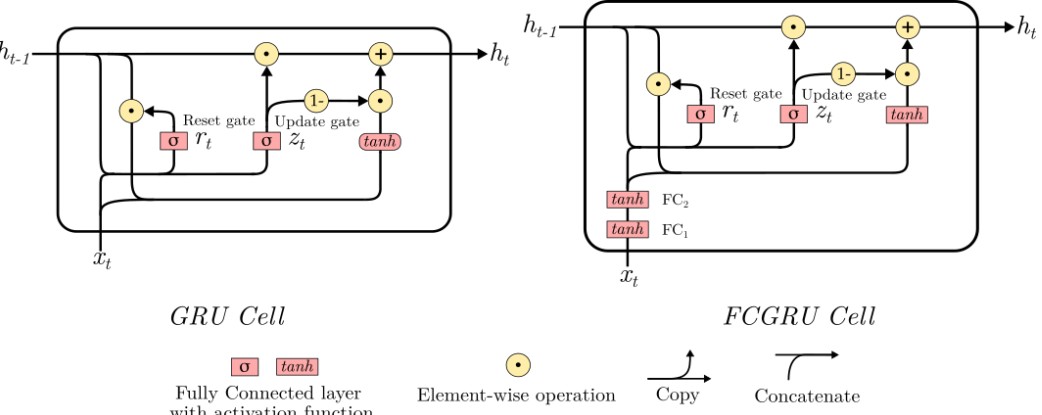

**Figure 2.** Visual representation of the GRU and FCGRU cells [17].

The classification model is trained via a semi-supervised procedure with the aim to leverage as much as possible the information available considering both labelled and unlabelled samples [30,31]. The procedure is depicted in Figure 3. More precisely, we considered a consistency-regularization-based strategy where the binary classification model is forced to behave similarly to an auxiliary model trained on the perturbed data. Such a consistency regularization strategy was deployed considering a reduced set of unlabelled samples $\tilde{U} = U \setminus U_{\mu_{rec}}$.

We denote by $f_\theta$ the binary classification model. When only labelled data $P$ and $RN$ are considered, the model is trained via the standard binary cross-entropy (BCE) loss, defined as:

$$Loss_{BCE}(Y, P \cup RN, f_\theta) = -\frac{1}{|P \cup RN|} \sum_{i=1}^{|P \cup RN|} y_i \log(f_\theta(TS_i)) + (1 - y_i) \log(f_\theta(1 - TS_i)) \tag{10}$$

where $TS_i \in P \cup RN$ is a generic satellite image time series sample with an associated label $y_i \in Y$ and $Y$ is the set of labels.

With the aim to exploit available unlabelled data $\tilde{U}$ in the learning process, we added an extra term to the main loss function as follows:

$$Loss = Loss_{BCE}(Y, P \cup RN, f_\theta) + \lambda \frac{1}{|\tilde{U}|} \sum_{j=1}^{|\tilde{U}|} KL(f_{\hat{\theta}}(\hat{TS}_j) || f_\theta(TS_j)) \tag{11}$$

where $TS_j \in \tilde{U}$ and $f_{\hat{\theta}}$ is an auxiliary classification model that has the same network structure as $f_\theta$, but trained on $\hat{P}$ and $\hat{RN}$, which are the reconstructed (via the VAERNN autoencoder) positive and reliable negative samples, respectively. $KL(X||Z)$ is the standard

Kullback–Leibler divergence, which quantifies how much the distribution $Z$ differs from the distribution $X$. The second term of the semi-supervised loss function has the goal of leveraging the unlabelled set of SITS data, $\tilde{U}$, regularizing the classification behaviour via consistency assessment [31]. This is achieved by imposing that the output distribution of the $f_\theta$ model should behave similarly to the output distribution of a perturbed version of itself, named $f_{\hat{\theta}}$, trained on the set of reconstructed positive and reliable negative samples. Such a strategy implicitly enforces the classifier $f_\theta$ to produce similar output distributions considering a specific sample and its perturbed version, thus stressing smoothness on the manifold on which the data are projected by the neural network model [31]. The trade-off hyper-parameter $\lambda$ is equal to 2.

To implement such a training strategy, at each iteration of an epoch, the classification models $f_\theta$ and $f_{\hat{\theta}}$ were trained simultaneously. Firstly, $f_{\hat{\theta}}$ was trained on a batch of samples coming from the perturbed data, and successively, the weight parameters of the $f_\theta$ model were updated considering the binary cross-entropy loss on a batch of labelled samples from $P \cup RN$ plus the consistency regularization loss on a batch of unlabelled samples from the set $\tilde{U}$.

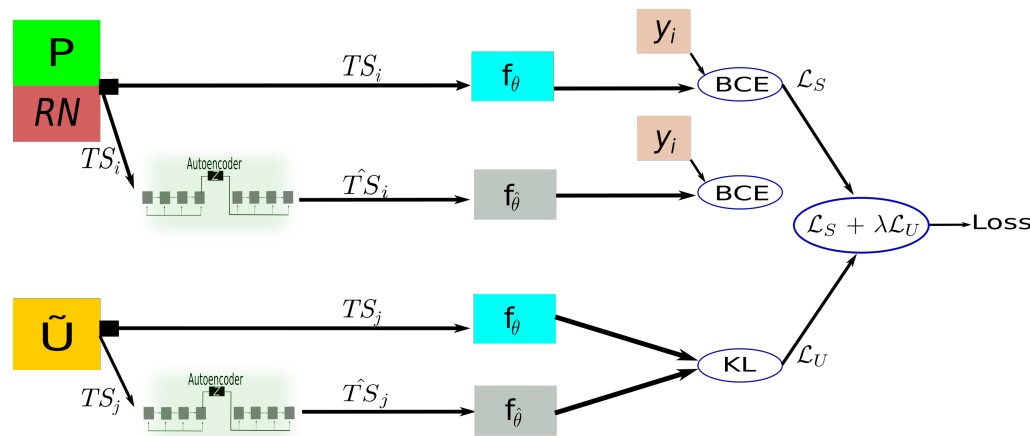

**Figure 3.** The training strategy involves two models trained simultaneously. The first model $f_{\hat{\theta}}$ is trained on the set of perturbed positive labelled $P$ and reliable negative $RN$ samples by the autoencoder with the supervised BCE loss on true labels $y_i$. The second $f_\theta$ is the final classification model trained on $P$ and $RN$ samples, with a semi-supervised process involving two losses: the binary cross entropy loss on the true labels $y_i$ ($\mathcal{L}_S$) and the KL divergence between the generated soft labels $f_\theta(TS_j)$ and $f_{\hat{\theta}}(\hat{TS}_j)$ to enforce consistency ($\mathcal{L}_U$). The $\lambda$ hyper-parameter weights the importance of the unsupervised term.

At the end of the training procedure, the classifier $f_\theta$ is the model that is employed to perform inference on new unseen samples.

## 4. Data

The study area we considered in our research study, namely *Haute-Garonne*, is located in the southwest area of France (Figure 4a), and it covers a surface area of 4146.2 km$^2$. The satellite data describing the study site consist of a Sentinel-2 (S2) time series involving 31 images acquired over the year 2019 from January to December. The temporal distribution of the satellite image time series is depicted in Figure 5.

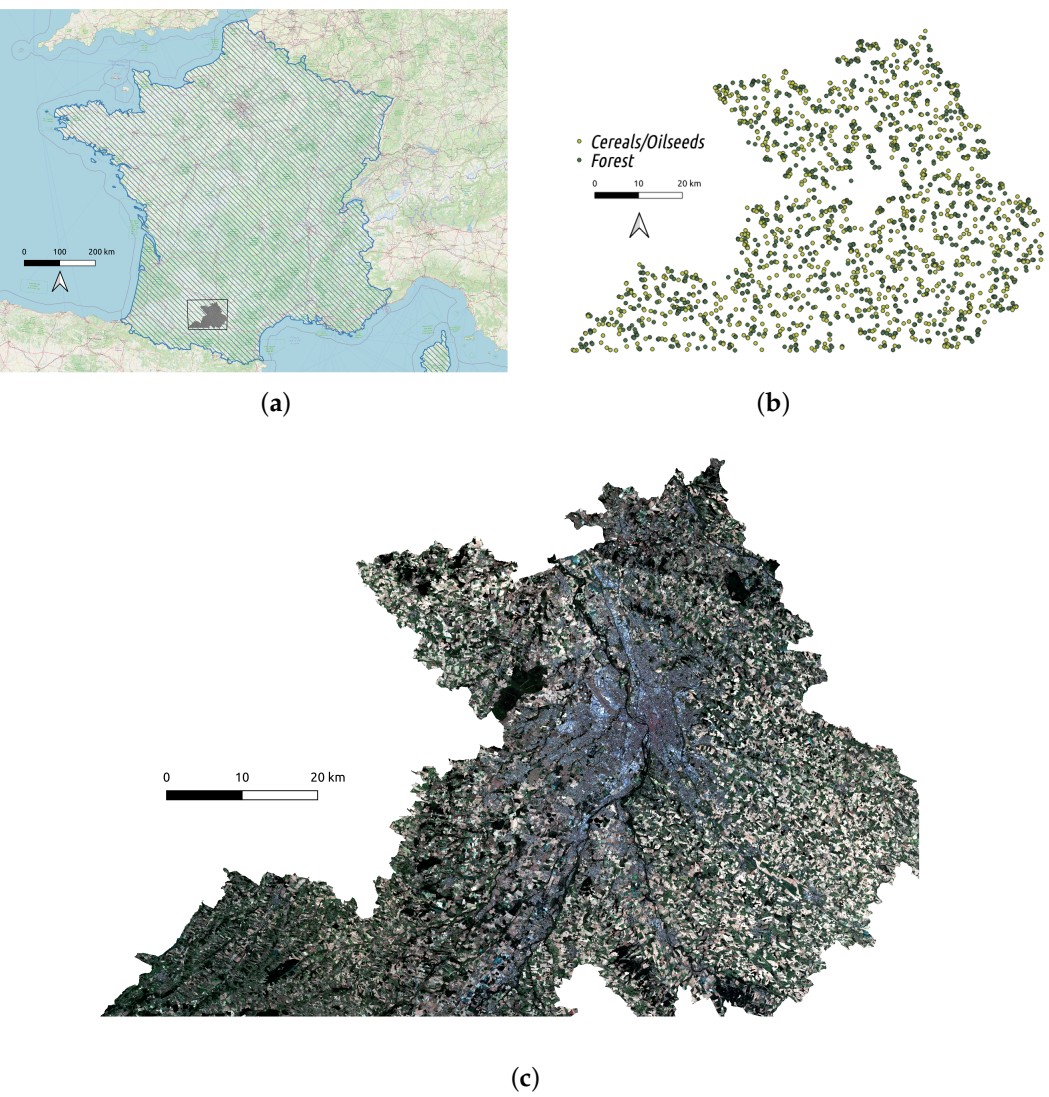

(a)　　　　　　　　　　　　　　　　　(b)

(c)

**Figure 4.** *Haute-Garonne* is located in the southwest of France (**a**). The spatial distribution of the land covers (*Cereals/Oilseeds* and *Forest*) considered as positive classes (**b**). The Sentinel-2 RGB composite obtained over the year 2019 (**c**).

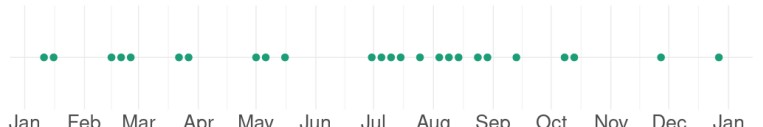

**Figure 5.** Overview of the acquisition dates of Sentinel-2 (S2) images over *Haute-Garonne*. S2 acquisitions are sparsely distributed due to the ubiquitous cloudiness.

The Sentinel-2 images were downloaded from the THEIA pole platform (http://theia. cnes.fr, accessed on 10 November 2020) at level-2A in top-of-canopy reflectance values and provided with cloud masks. Only 10 m spatial resolution bands (blue, green, red, and near-infrared spectrum) were considered in this analysis. Additionally, two common spectral indices were then extracted: the NDVI [32] and the NDWI [33]. A preprocessing step was performed over each band to replace cloudy pixel values as detected by the available cloud masks through a linear multi-temporal interpolation (cf. temporal gap-filling [5]). The entire study site is enclosed in the Sentinel-2 tile T31TCJ.

**Table 1.** *Haute-Garonne* ground truth characteristics.

| Class | # of Objects | # of Pixels |
|---|---|---|
| *Cereals/Oilseeds* | 898 | 336,044 |
| *Forest* | 846 | 186,870 |
| Other land cover classes | 6460 | 323,925 |
| Total | 7358 | 846,838 |

Since we are addressing a positive and unlabelled learning setting, we considered two different scenarios where each scenario involves a particular land cover class as the positive class and all the other land cover classes as negative. As positive classes, we considered the *Cereals/Oilseeds* and the *Forest* land covers.

The statistics about the *Cereals/Oilseeds* and *Forest* land cover classes, in the *Haute-Garonne* study site, are reported in Table 1. We also report the statistics for the number of labelled samples belonging to the other land cover classes present in the *Haute-Garonne* study area (i.e., Water, Urban, Orchards, etc.) that are neither *Cereals/Oilseeds* nor *Forest* samples.

The ground truth (GT) was obtained from various sources: (i) the *Registre Parcellaire Graphique* (RPG) reference data for 2019 (https://www.data.gouv.fr/en/datasets, accessed on 10 November 2020), which is part of the European Land Parcel Identification System (LPIS) provided by the French Agency for services and payment ; (ii) the French Topographic Database (BD TOPO) (https://geoservices.ign.fr/blog/2020/10/26/Grand_angle_diffusion_BDTOPO.html, accessed on 10 November 2020), which contains geographical and administrative entities of the national territory; (iii) the French National Forest inventory (BD FORET), which includes information about forest cover at the national scale. The GT was assembled in a Geographic Information System (GIS) vector file, containing a collection of polygons, each attributed to a land cover category (see Table 1). Finally, the polygons were rasterized at the Sentinel-2 spatial resolution (10 m).

## 5. Experimental Evaluation

In this section, we describe the experimental evaluation we conducted with the aim to assess the behaviour of our framework. To this end, we firstly compared PUL-SITS with different competing approaches on the study site introduced in Section 4. Secondly, we evaluated the different components on which PUL-SITS is based on through an ablation study. Finally, we visually inspected the land cover maps produced by the different competing approaches.

### 5.1. Competing Methods and Ablation

To assess the behaviour of PUL-SITS, we considered the following competitors:

- A random forest approach based on the positive unlabelled learning framework introduced in [34]. Here, the authors proposed to exploit probability theory to weight unlabelled samples with the aim to bias the learning stage of standard supervised classification methods. We named such a competitor $RF_{PUL}$;
- A one-class support vector machine approach [35]. In this case, the learning approach discards unlabelled samples, and it only leverages positive samples to build the final binary classification model. We named such a competing approach OCSVM;
- An ensemble-based approach for the positive unlabelled learning scenario recently introduced in the field of remote sensing analysis [14]. The ensemble approach combines the prediction of different base learners via a mean-squared-error-based aggregation function. Our ensemble-based competitor involves the following base learners: logistic ridge regression, logistic elastic-net regression, random forest, multilayer perceptron and linear discriminant analysis. We named such competitors $ENSEMBLE_{PUL}$.

In order to disentangle the contributions of the different stages on which PUL-SITS is based, we considered the following ablations of our framework:

- The FCGRU [17] classification model only trained on the positive and reliable negative samples obtained from the first stage of our framework. This ablation did not involve the consistency loss. We named such an ablation PUL-SITS$_{noReg}$;
- The FCGRU classification model trained on the positive and reliable negative samples obtained from the first stage of our framework. Here, the classifier was not trained on the original data, but on the reconstructed data output by the autoencoder model. We named such an ablation PUL-SITS$_{reco}$.

### 5.2. Experimental Settings

We designed two different positive unlabelled learning experiments according to the data presented in Section 4. We remind the reader that in the first (resp. second) experiment, we considered as the positive class the *Cereals/Oilseeds* (resp. *Forest*) land cover class, while the negative class was constituted by all the samples not belonging to the land cover class representing the positive class.

With the purpose of setting up a fair evaluation protocol to assess the quality of the different competitors, for each experiment, we divided the data (both positive and negative classes) into two sets: training and test. We considered 50% of the positive (resp. negative) class as the test data. The remaining 50% of the data were considered as the training data. Then, the training set was split again into two parts: the positive and the unlabelled set. While the former contained only positive samples, the latter consisted of samples from both the positive and negative classes. Whereas the amount of positive samples can influence the model behaviour, we considered increasing the quantity of positive samples ranging in the set $\{20, 40, 60, 80, 100\}$ in terms of objects. Once the set of positive samples was fixed, the rest of the training data were considered as unlabelled data. The number of objects and pixels for the positive, unlabelled, and negative samples in the training and test sets for the first (*Cereals/Oilseeds* land cover as the positive class) and second (*Forest* land cover as the positive class) experiments is reported in Table 2 (resp. Table 3).

**Table 2.** Number of positive, unlabelled, and negative objects/pixels in the training and test set when the positive samples belong to the *Cereals/Oilseeds* land cover class.

| # of Objects Training (P/U) | # of Pixels Training (P/U) | # of Objects Test (P/N) | # of Pixels Test (P/N) |
|---|---|---|---|
| 20/3659 | 7010/410,712 | | |
| 40/3639 | 14,618/403,104 | | |
| 60/3619 | 21,001/403,822 | 451/3228 | 168,404/260,711 |
| 80/3599 | 28,955/388,767 | | |
| 100/3579 | 36,442/381,280 | | |

**Table 3.** Number of positive, unlabelled, and negative objects/pixels in the training and test set when the positive samples belong to the *Forest* land cover class.

| # of Objects Train (P/U) | # of Pixels Train (P/U) | # of Objects Test (P/N) | # of Pixels Test (P/N) |
|---|---|---|---|
| 20/3658 | 4206/413,517 | | |
| 40/3638 | 9565/408,157 | | |
| 60/3618 | 13,900/403,822 | 428/3250 | 95,725/333,390 |
| 80/3598 | 17,865/399,857 | | |
| 100/3578 | 22,143/395,579 | | |

We remind the reader that, despite having defined the size of the positive samples in terms of objects, the classification procedure was performed at the pixel level (all the pixels belonging to the object belong exclusively to the training or the test set). We organized the

training/test data split procedures in terms of objects with the aim to avoid possible spatial bias in the evaluation procedure [5], since, in this way, pixels belonging to the same ground truth object were assigned exclusively to one of the data partitions. In addition, for time series data, all values were normalized per band in the interval $[0, 1]$ using 2% (resp. 98%), rather than the minimum (resp. maximum) value to rescale the data [8].

Regarding our framework, PUL-SITS, the VAERNN involves an encoder with two levels with 64 and 16 units, respectively, while the decoder is symmetric to the encoder. The FCGRU [17] recurrent cell involved in the binary classification process has a number of units equal to 32. Both models were trained for 50 epochs with a batch size equal to 32 and a learning rate of $10e^{-4}$ through the Adam optimizer. For the latter component, we also employed a dropout rate equal to 0.2. The same learning procedure was employed for the PUL-SITS$_{noReg}$ and PUL-SITS$_{reco}$ ablations. Regarding the competing methods, RF$_{PUL}$, OCSVM, and ENSEMBLE$_{PUL}$, they were employed with the default settings.

The assessment of the quantitative performances was performed considering the *Accuracy*, *F-Measure*, *Sensitivity*, *Specificity*, and *Kappa* metrics [36]. To reduce the bias induced by the training/test data split, all the results were averaged over ten different random splits. Finally, both the average and standard deviation are reported for each evaluation metric.

The experiments were carried out on a workstation with an AMD Ryzen 7 3700X 8-core processor CPU@3.60GHz with 64 GB of RAM and a GeForce RTX 2080 Ti GPU. All the deep-learning methods were implemented using the Python Tensorflow library.

### 5.3. Quantitative Results

As described before, we compared PUL-SITS with respect to the competing methods on two different experiments: the first one involved the *Cereals/Oilseeds* land cover as the class of interest, while the second experiment considered the *Forest* land cover as the positive class. In both cases, we firstly assessed the performances of the competing strategies in terms of the standard evaluation metrics, then we discuss the associate confusion matrices; finally, we provide a qualitative analysis, inspecting some extracts of the produced land cover maps.

### 5.3.1. Experiments with the *Cereals/Oilseeds* Land Cover as the Positive Class

Tables 4–8 summarise the results in terms of the F-Measure, Accuracy, Sensitivity, Specificity, and Kappa measure, respectively. Generally, we can observe that no matter the size of the training data, our framework always outperformed all the competing approaches considering almost all the evaluation metrics. We can also note that all the approaches exhibited better behaviours as the amount of available positive labelled data increased.

Regarding the competing methods, we can observe that OCSVM achieved the best performance. Such a method, as opposed to RF$_{PUL}$ and ENSEMBLE$_{PUL}$, only exploited the positive samples to build its classification model. This fact underlines that RF$_{PUL}$ and ENSEMBLE$_{PUL}$ are probably impacted by the fact that the negative class (all the land cover classes except *Cereals/Oilseeds*) is characterised by a high intra-heterogeneity, thus influencing the ability of the algorithm to model such a wide range of sample as one class in the original feature space. This fact is illustrated by high Sensitivity (Table 6) and low Specificity (Table 7) values, which highlight that RF$_{PUL}$ and ENSEMBLE$_{PUL}$ simply label the majority of the data points with the negative class. Concerning the ablations associated with our framework, PUL-SITS$_{noReg}$ and PUL-SITS$_{reco}$, we can see that both strategies achieved better performances than the competing methods, with PUL-SITS$_{noReg}$ as a clear winner between them, but the PUL-SITS$_{reco}$ component played an important role in supporting the performance of PUL-SITS, underlying the quality of our two-stage methodology. Figure 6 shows the confusion matrices associated with the three direct competing approaches, as well as PUL-SITS. A closer look at these statistics confirms that PUL-SITS was more precise than the competitors. This consideration emerges from the fact that the corresponding heat map (Figure 6d) has a more visible diagonal structure (the

dark green blocks concentrated on the diagonal). This is not the case for the other methods, where the majority of the samples are associated with the negative (zero) class.

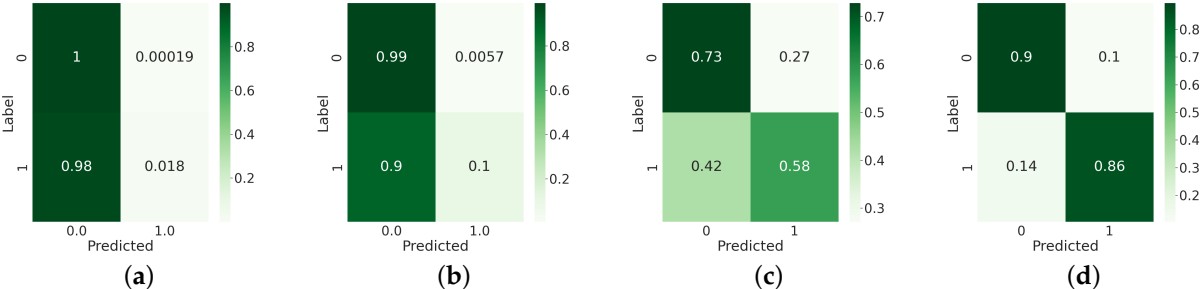

**Figure 6.** Confusion matrices of (**a**) RF$_{PUL}$, (**b**) ENSEMBLE$_{PUL}$, (**c**) OCSVM, and (**d**) PUL-SITS considering the case in which the land cover class of interest (the positive class) is *Cereals/Oilseeds* with the number of positive training data objects (*n*) equal to 80.

**Table 4.** Performances in terms of the F-Measure of the different competing strategies when the land cover class of interest (the positive class) is *Cereals/Oilseeds*.

| # of Obj Samples | RF$_{PUL}$ | ENSEMBLE$_{PUL}$ | OCSVM | PUL-SITS$_{noReg}$ | PUL-SITS$_{reco}$ | PUL-SITS |
|---|---|---|---|---|---|---|
| 20 | 46.2 ± 1.7 | 47.3 ± 1.5 | 61.6 ± 4.0 | **80.3** ± 2.3 | 75.5 ± 3.5 | 78.9 ± 4.0 |
| 40 | 46.7 ± 1.7 | 49.3 ± 1.7 | 63.8 ± 2.7 | 84.9 ± 1.9 | 82.0 ± 2.3 | **85.3** ± 2.8 |
| 60 | 47.2 ± 1.8 | 52.4 ± 1.6 | 64.7 ± 2.0 | 86.6 ± 2.8 | 84.2 ± 2.0 | **87.3** ± 2.2 |
| 80 | 48.3 ± 2.0 | 55.7 ± 2.1 | 64.7 ± 2.3 | 86.5 ± 2.3 | 85.0 ± 2.5 | **88.0** ± 2.6 |
| 100 | 50.2 ± 1.6 | 60.1 ± 1.8 | 64.3 ± 2.0 | 86.2 ± 3.0 | 85.5 ± 1.8 | **88.6** ± 1.3 |

**Table 5.** Performances in terms of the Accuracy of the different competing strategies when the land cover class of interest (the positive class) is *Cereals/Oilseeds*.

| # of Obj Samples | RF$_{PUL}$ | ENSEMBLE$_{PUL}$ | OCSVM | PUL-SITS$_{noReg}$ | PUL-SITS$_{reco}$ | PUL-SITS |
|---|---|---|---|---|---|---|
| 20 | 60.9 ± 1.4 | 61.3 ± 1.3 | 63.9 ± 2.1 | **80.9** ± 2.0 | 76.3 ± 3.1 | 79.5 ± 3.6 |
| 40 | 61.1 ± 1.4 | 62.1 ± 1.2 | 65.1 ± 2.2 | 85.0 ± 1.9 | 82.3 ± 2.2 | **85.4** ± 2.8 |
| 60 | 61.3 ± 1.4 | 63.5 ± 1.2 | 66.0 ± 1.6 | 86.6 ± 2.8 | 84.2 ± 2.0 | **87.3** ± 2.1 |
| 80 | 61.8 ± 1.5 | 65.1 ± 1.3 | 65.4 ± 1.9 | 86.5 ± 2.3 | 85.1 ± 2.4 | **87.9** ± 2.5 |
| 100 | 62.6 ± 1.3 | 67.5 ± 1.2 | 65.2 ± 1.7 | 86.1 ± 2.9 | 85.5 ± 1.8 | **88.6** ± 1.3 |

**Table 6.** Performances in terms of the Sensitivity of the different competing strategies when the land cover class of interest (the positive class) is *Cereals/Oilseeds*.

| # of Obj Samples | RF$_{PUL}$ | ENSEMBLE$_{PUL}$ | OCSVM | PUL-SITS$_{noReg}$ | PUL-SITS$_{reco}$ | PUL-SITS |
|---|---|---|---|---|---|---|
| 20 | **100** ± 0.0 | 99.8 ± 0.1 | 81.5 ± 5.0 | 91.7 ± 4.1 | 88.1 ± 3.5 | 88.6 ± 8.9 |
| 40 | **100** ± 0.0 | 99.6 ± 0.2 | 79.6 ± 3.9 | 88.4 ± 2.8 | 89.0 ± 1.8 | 90.3 ± 2.0 |
| 60 | **100** ± 0.0 | 99.3 ± 0.3 | 77.3 ± 3.8 | 86.7 ± 5.8 | 87.3 ± 3.6 | 88.3 ± 2.5 |
| 80 | **100** ± 0.0 | 99.1 ± 0.2 | 77.0 ± 3.6 | 86.6 ± 5.3 | 88.9 ± 2.5 | 88.7 ± 2.0 |
| 100 | **99.9** ± 0.1 | 98.9 ± 0.2 | 77.7 ± 3.2 | 85.1 ± 5.9 | 87.4 ± 2.8 | 89.4 ± 2.0 |

**Table 7.** Performances in term of the Specificity of the different competing strategies when the land cover class of interest (the positive class) is *Cereals/Oilseeds*.

| # of Obj Samples | $RF_{PUL}$ | $ENSEMBLE_{PUL}$ | OCSVM | $PUL\text{-}SITS_{noReg}$ | $PUL\text{-}SITS_{reco}$ | PUL-SITS |
|---|---|---|---|---|---|---|
| 20 | $0.3 \pm 0.3$ | $1.6 \pm 0.8$ | $36.9 \pm 11.2$ | $65.3 \pm 9.8$ | $59.3 \pm 10.0$ | $\mathbf{66.6 \pm 13.6}$ |
| 40 | $0.9 \pm 0.6$ | $4.0 \pm 2.2$ | $42.7 \pm 7.7$ | $\mathbf{80.0 \pm 4.7}$ | $72.4 \pm 5.2$ | $78.2 \pm 5.3$ |
| 60 | $1.4 \pm 0.7$ | $8.1 \pm 2.1$ | $47.4 \pm 6.8$ | $85.9 \pm 2.9$ | $79.6 \pm 5.0$ | $\mathbf{85.9 \pm 5.0}$ |
| 80 | $2.7 \pm 0.9$ | $12.4 \pm 3.0$ | $47.7 \pm 6.9$ | $86.1 \pm 4.2$ | $79.4 \pm 5.5$ | $\mathbf{86.8 \pm 4.5}$ |
| 100 | $5.0 \pm 1.2$ | $18.8 \pm 2.8$ | $45.9 \pm 6.0$ | $\mathbf{87.6 \pm 3.1}$ | $82.7 \pm 3.7$ | $87.3 \pm 4.3$ |

**Table 8.** Performances in term of the Kappa measure of the different competing strategies when the land cover class of interest (the positive class) is *Cereals/Oilseeds*.

| # of Obj Samples | $RF_{PUL}$ | $ENSEMBLE_{PUL}$ | OCSVM | $PUL\text{-}SITS_{noReg}$ | $PUL\text{-}SITS_{reco}$ | PUL-SITS |
|---|---|---|---|---|---|---|
| 20 | $0.00 \pm 0.00$ | $0.02 \pm 0.01$ | $0.19 \pm 0.07$ | $\mathbf{0.59 \pm 0.05}$ | $0.49 \pm 0.07$ | $0.56 \pm 0.07$ |
| 40 | $0.01 \pm 0.01$ | $0.04 \pm 0.02$ | $0.23 \pm 0.05$ | $0.68 \pm 0.04$ | $0.62 \pm 0.05$ | $\mathbf{0.69 \pm 0.06}$ |
| 60 | $0.02 \pm 0.01$ | $0.09 \pm 0.02$ | $0.25 \pm 0.04$ | $0.72 \pm 0.05$ | $0.67 \pm 0.04$ | $\mathbf{0.74 \pm 0.04}$ |
| 80 | $0.03 \pm 0.01$ | $0.14 \pm 0.03$ | $0.25 \pm 0.04$ | $0.72 \pm 0.04$ | $0.69 \pm 0.05$ | $\mathbf{0.75 \pm 0.05}$ |
| 100 | $0.06 \pm 0.01$ | $0.21 \pm 0.03$ | $0.24 \pm 0.04$ | $0.72 \pm 0.05$ | $0.70 \pm 0.03$ | $\mathbf{0.76 \pm 0.03}$ |

### 5.3.2. Experiments with the *Forest* Land Cover as the Positive Class

Tables 9–13 resume respectively the results in terms of the F-Measure, Accuracy, Sensitivity, Specificity, and Kappa of the different competing methods when the *Forest* land cover was considered as the positive class. In this case also, PUL-SITS outperformed all the other approaches, while $RF_{PUL}$ and $ENSEMBLE_{PUL}$ simply labelled all of the data points as the negative class (as highlighted by the high values of the Sensitivity and the low values of the Sensibility they achieved). OCSVM exhibited the best performance among the state-of-the-art competitors. Concerning the ablations, $PUL\text{-}SITS_{noReg}$ did not always achieve better performance over OCSVM, while $PUL\text{-}SITS_{reco}$ did. This phenomena clearly underlines the importance of exploiting the available unlabelled data and the appropriateness of the consistency-based procedure in order to regularize the training stage of the final classification model. Figure 7 shows the confusion matrices, which confirm that PUL-SITS achieved a better predictive performance w.r.t. the other approaches, i.e., it has a more visible diagonal structure.

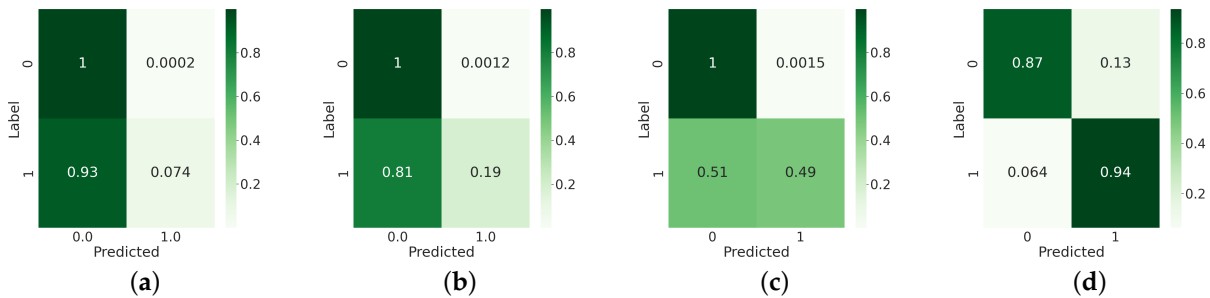

**Figure 7.** Confusion matrices of (**a**) $RF_{PUL}$, (**b**) $ENSEMBLE_{PUL}$, (**c**) OCSVM, and (**d**) PUL-SITS considering the case in which the land cover class of interest (the positive class) is *Forest* with the number of positive training objects (*n*) equal to 80.

**Table 9.** Performances in terms of the F-Measure of the different competing strategies when the land cover class of interest (the positive class) is *Forest*.

| # of Obj Samples | RF$_{PUL}$ | ENSEMBLE$_{PUL}$ | OCSVM | PUL-SITS$_{noReg}$ | PUL-SITS$_{reco}$ | PUL-SITS |
|---|---|---|---|---|---|---|
| 20 | 68.4 ± 2.7 | 69.8 ± 2.4 | 82.7 ± 3.0 | 86.1 ± 1.8 | 89.7 ± 2.0 | **91.4** ± 1.9 |
| 40 | 69.7 ± 2.7 | 72.8 ± 3.1 | 83.1 ± 2.6 | 85.1 ± 3.4 | 89.8 ± 3.7 | **92.9** ± 2.3 |
| 60 | 71.6 ± 2.9 | 76.4 ± 3.1 | 84.4 ± 2.5 | 86.5 ± 2.5 | 90.1 ± 2.2 | **92.6** ± 2.5 |
| 80 | 73.9 ± 3 | 79.5 ± 3.1 | 84.7 ± 2.4 | 86.6 ± 2.9 | 88.8 ± 2.9 | **91.4** ± 3.5 |
| 100 | 76.8 ± 2.7 | 83.1 ± 2.8 | 85.2 ± 2.4 | 85.0 ± 1.7 | 88.7 ± 2.8 | **91.9** ± 1.8 |

**Table 10.** Performances in terms of the Accuracy of the different competing strategies when the land cover class of interest (the positive class) is *Forest*.

| # of Obj Samples | RF$_{PUL}$ | ENSEMBLE$_{PUL}$ | OCSVM | PUL-SITS$_{noReg}$ | PUL-SITS$_{reco}$ | PUL-SITS |
|---|---|---|---|---|---|---|
| 20 | 77.9 ± 1.9 | 78.4 ± 1.8 | 85.0 ± 2.1 | 83.2 ± 2.3 | 87.9 ± 2.6 | **90.1** ± 2.7 |
| 40 | 78.4 ± 1.9 | 79.8 ± 2.1 | 85.8 ± 1.9 | 81.9 ± 4.6 | 88.0 ± 4.8 | **92.1** ± 2.8 |
| 60 | 79.3 ± 2.0 | 81.7 ± 2.1 | 86.7 ± 1.8 | 83.8 ± 3.8 | 88.4 ± 2.9 | **91.5** ± 3.1 |
| 80 | 80.4 ± 2.0 | 83.5 ± 2.2 | 86.9 ± 1.8 | 83.7 ± 3.9 | 86.7 ± 3.7 | **90.1** ± 4.3 |
| 100 | 81.9 ± 1.9 | 85.8 ± 2.1 | 87.2 ± 1.8 | 81.8 ± 2.3 | 86.7 ± 3.7 | **90.8** ± 2.2 |

**Table 11.** Performances in terms of the Sensitivity of the different competing strategies when the land cover class of interest (the positive class) is *Forest*.

| # of Obj Samples | RF$_{PUL}$ | ENSEMBLE$_{PUL}$ | OCSVM | PUL-SITS$_{noReg}$ | PUL-SITS$_{reco}$ | PUL-SITS |
|---|---|---|---|---|---|---|
| 20 | **100** ± 0.0 | 99.9 ± 0.1 | 99.7 ± 0.2 | 81.6 ± 2.5 | 87.2 ± 3.0 | 89.7 ± 2.9 |
| 40 | **100** ± 0.0 | 99.8 ± 0.1 | 99.7 ± 0.3 | 80.1 ± 5.1 | 87.1 ± 5.5 | 91.9 ± 3.3 |
| 60 | **99.9** ± 0.0 | 99.8 ± 0.2 | 99.7 ± 0.2 | 82.0 ± 3.7 | 87.6 ± 3.4 | 91.2 ± 3.6 |
| 80 | **99.9** ± 0.1 | 99.8 ± 0.1 | 99.7 ± 0.2 | 82.2 ± 4.3 | 85.7 ± 4.3 | 89.5 ± 5.0 |
| 100 | **99.8** ± 0.1 | 99.7 ± 0.1 | 99.7 ± 0.1 | 79.9 ± 2.6 | 85.6 ± 4.3 | 90.2 ± 2.6 |

**Table 12.** Performances in terms of the Specificity of the different competing strategies when the land cover class of interest (the positive class) is *Forest*.

| # of Obj Samples | RF$_{PUL}$ | ENSEMBLE$_{PUL}$ | OCSVM | PUL-SITS$_{noReg}$ | PUL-SITS$_{reco}$ | PUL-SITS |
|---|---|---|---|---|---|---|
| 20 | 0.8 ± 0.4 | 3.7 ± 1.3 | 36.0 ± 7.8 | **97.3** ± 1.01 | 94.1 ± 1.5 | 94.0 ± 1.6 |
| 40 | 3.4 ± 2.4 | 10.1 ± 3.7 | 37.2 ± 7.2 | **97.6** ± 1.26 | 95.2 ± 1.8 | 93.6 ± 2.5 |
| 60 | 7.5 ± 3.2 | 18.7 ± 5.2 | 41.1 ± 6.9 | **97.9** ± 1.0 | 95.5 ± 1.7 | 94.3 ± 2.9 |
| 80 | 12.5 ± 4.2 | 26.8 ± 6.3 | 42.0 ± 6.5 | **97.9** ± 1.0 | 96.2 ± 1.5 | 95.2 ± 2.2 |
| 100 | 19.6 ± 4.7 | 37.3 ± 6.8 | 43.6 ± 6.5 | **98.3** ± 0.7 | 96.3 ± 1.4 | 94.9 ± 2.0 |

**Table 13.** Performances in terms of the Kappa measure of the different competing strategies when the land cover class of interest (the positive class) is *Forest*.

| # of Obj Samples | RF$_{PUL}$ | ENSEMBLE$_{PUL}$ | OCSVM | PUL-SITS$_{noReg}$ | PUL-SITS$_{reco}$ | PUL-SITS |
|---|---|---|---|---|---|---|
| 20 | 0.01 ± 0.01 | 0.05 ± 0.02 | 0.46 ± 0.09 | 0.46 ± 0.05 | 0.55 ± 0.07 | **0.61** ± 0.08 |
| 40 | 0.05 ± 0.04 | 0.14 ± 0.05 | 0.47 ± 0.07 | 0.44 ± 0.08 | 0.57 ± 0.13 | **0.67** ± 0.09 |
| 60 | 0.11 ± 0.05 | 0.26 ± 0.07 | 0.51 ± 0.07 | 0.47 ± 0.08 | 0.57 ± 0.08 | **0.66** ± 0.1 |
| 80 | 0.18 ± 0.06 | 0.36 ± 0.08 | 0.52 ± 0.07 | 0.48 ± 0.08 | 0.54 ± 0.08 | **0.62** ± 0.1 |
| 100 | 0.27 ± 0.06 | 0.47 ± 0.08 | 0.54 ± 0.07 | 0.44 ± 0.05 | 0.53 ± 0.07 | **0.63** ± 0.05 |

### 5.4. Visual Inspection of the Binary Classification Maps

Here, we performed a visual inspection of the binary classification maps generated via the different competing approaches regarding the two scenarios we previously described.

Figures 8 and 9 report the results of the mapping process on two different extracts of the study site when the *Cereals/Oilseeds* land cover class was considered as the positive class. The first extract (Figure 8) depicts an agricultural area mainly characterized by cereal and oilseed parcels. We can observe that PUL-SITS clearly recovered the parcel structure, while the other approaches had serious issues in detecting pixels belonging to the *Cereals/Oilseeds* class. While $RF_{PUL}$ and $ENSEMBLE_{PUL}$ exhibited high false-negative rates, the OCSVM method showed reasonable performance on the considered extract, but still (visually) inferior w.r.t. our framework. A similar behaviour was exhibited by all the competing methods on the second extract depicted in Figure 9. In this case also, PUL-SITS provided a more coherent land cover map that clearly discriminated between positive (*Cereals/Oilseeds*) and negative pixels.

Figures 10 and 11 depict the results of the mapping process on two different extracts of the study site when the *Forest* land cover class was considered as the positive class. The first extract (Figure 8) shows a portion of the study site where clear forest cover is present. Here, we can note that PUL-SITS well identified the large forest area in the bottom left part of the extract. All the other methods were not capable of recovering all the forest extent with, also in this case, issues related to false-negative detection. The second extract (Figure 11) describes a zone on which the forest cover is not concentrated, but, conversely, spread over all the area. Similar to what happened in the previous case, PUL-SITS produced a binary land cover map that was consistent with what we can observe in the RGB composite on the left side of the figure. To sum up, we can state that the visual inspection of the binary land cover maps, considering both scenarios, is coherent with the quantitative results we obtained in the previous stage of the evaluation analysis.

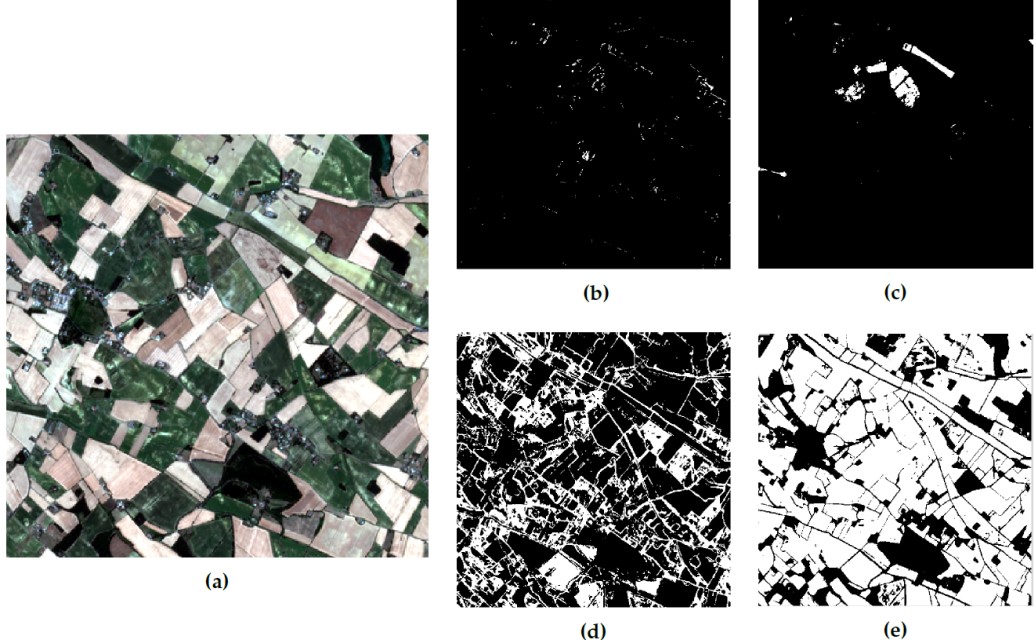

**Figure 8.** A first visual evaluation zone with (**a**) its RGB composite and the corresponding binary classification maps for (**b**) $RF_{PUL}$, (**c**) $ENSEMBLE_{PUL}$, (**d**) OCSVM, and (**e**) PUL-SITS for the positive class *Cereals/Oilseeds* with the number of positive training data objects (*n*) equal to 80.

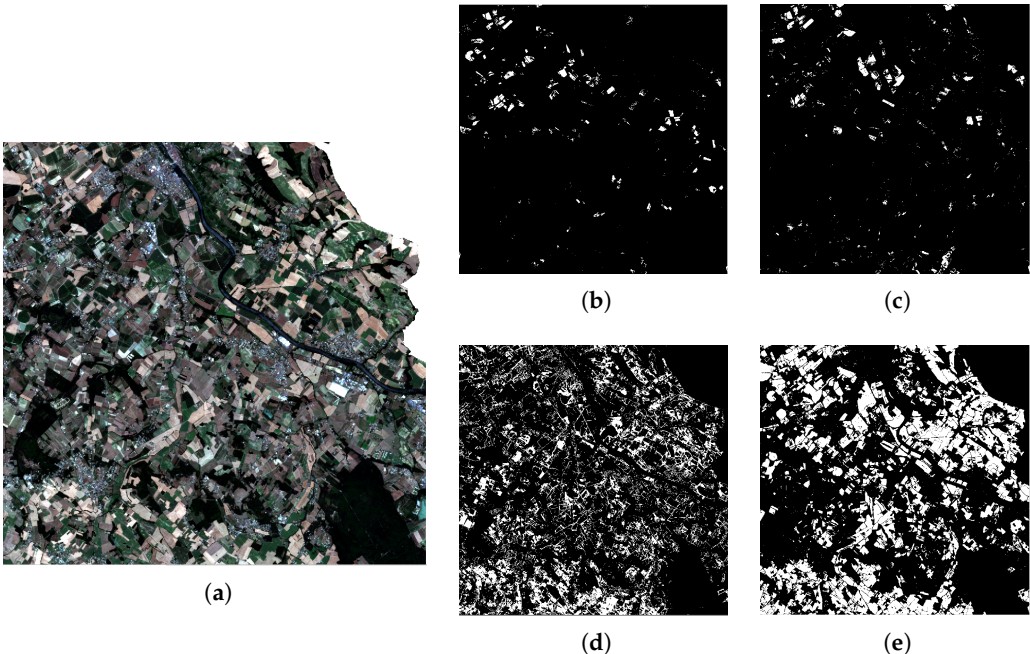

**Figure 9.** A second visual evaluation zone with (**a**) its RGB composite and the corresponding binary classification maps for (**b**) RF$_{PUL}$, (**c**) ENSEMBLE$_{PUL}$, (**d**) OCSVM, and (**e**) PUL-SITS for the positive class *Cereals/Oilseeds* with the number of positive training data objects (*n*) equal to 80.

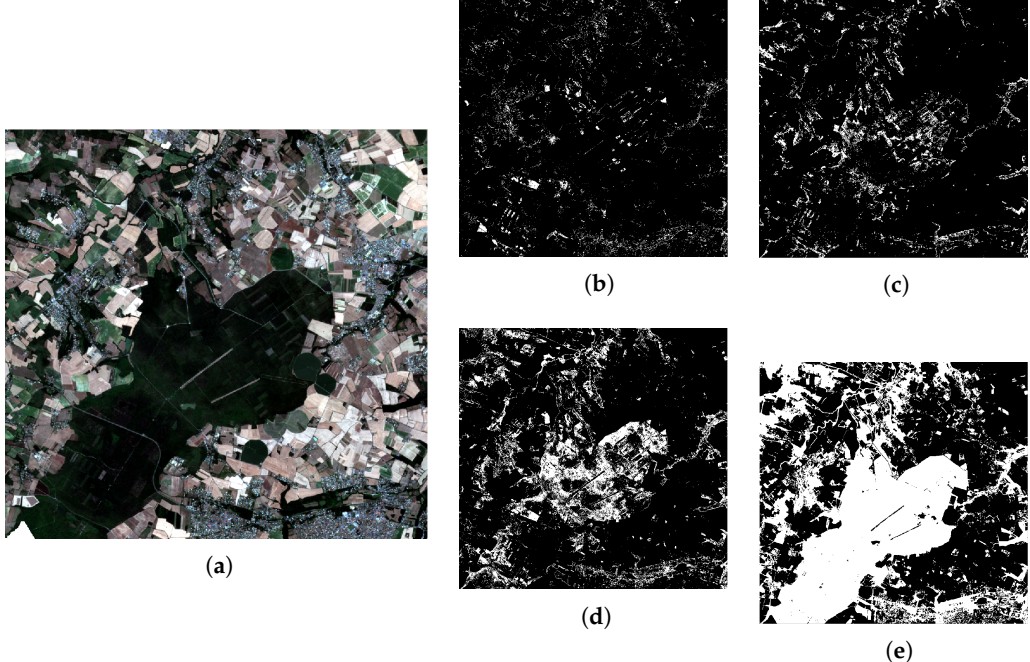

**Figure 10.** A first visual evaluation zone with (**a**) its RGB composite and the corresponding binary classification maps for (**b**) RF$_{PUL}$, (**c**) ENSEMBLE$_{PUL}$, (**d**) OCSVM, and (**e**) PUL-SITS for the positive class *Forest* with the number of positive training data objects (*n*) equal to 80.

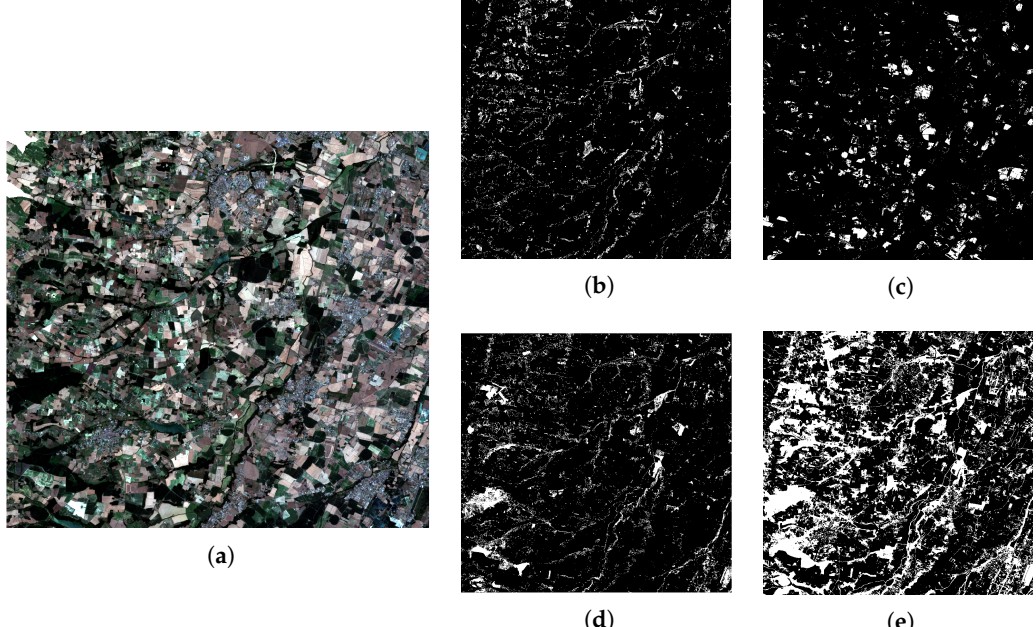

**Figure 11.** A second visual evaluation zone with (**a**) its RGB composite and the corresponding binary classification maps for (**b**) RF$_{PUL}$, (**c**) ENSEMBLE$_{PUL}$, (**d**) OCSVM, and (**e**) PUL-SITS for the positive class *Forest* with the number of positive training data objects (*n*) equal to 80.

## 6. Conclusions

In this research work, we introduced a novel framework, namely PUL-SITS, to deal with the positive unlabelled learning analysis of satellite image time series data for the task of land cover mapping. Our approach was based on a two-step strategy: in the first one, a recurrent neural network autoencoder was trained only on positive samples. Successively, the same autoencoder model was employed to filter out reliable negative samples. In the second stage, both labelled (positive and reliable negative) and unlabelled samples were exploited in a semi-supervised manner to build the final binary classification model. To assess the performances of PUL-SITS, we compared it with state-of-the-art methods for positive unlabelled learning considering two different scenarios: a first one in which the *Cereal/Oilseeds* land cover class has the role of the positive class while the rest of the land cover classes play the role of the negative class and a second scenario where the *Forest* land cover class was targeted as the positive class. Both scenarios underlined the quality of our approach. As future work, we plan to extend our framework towards multi-source analysis and introduce, for instance, satellite image time series of radar sensors (i.e., Sentinel-1) in order to exploit the complementarity between different available sources/modalities.

**Author Contributions:** D.I. and J.D. conceived and designed the experiments; J.D. performed the experiments; J.D. and D.I. analyzed the results; D.I. and J.D. wrote the article; D.I., A.B. and N.R. revised the paper. All authors have read and agreed to the published version of the manuscript.

**Funding:** This research was funded by ANRT, grant number CIFRE 2019/1993.

**Institutional Review Board Statement:** Not applicable.

**Informed Consent Statement:** Not applicable.

**Data Availability Statement:** Not applicable.

**Acknowledgments:** The authors wish also to thank the European Space Agency (ESA) for the Sentinel-2 data and Theia pole for the calibration of the Sentinel-2 images.

**Conflicts of Interest:** The authors declare no conflict of interest. The funders had no role in the design of the study; in the collection, analyses, or interpretation of data; in the writing of the manuscript, or in the decision to publish the results.

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
