# Peer review of "Positive Unlabelled Learning for Satellite Images’Time Series Analysis: An Application to Cereal and Forest Mapping"

_remotesensing, doi:10.3390/rs14010140_

Round 1

Reviewer 1 Report

Major revision.

This manuscript proposed a positive unlabelled learning method for satellite images. However, experimental analysis can not explain the effectiveness of the method. Experiments need to be re implemented to illustrate the effectiveness of the method.

Generally, the result accuracy of classical method is not very poor. Why the accuracy of the results of RF and ENSEMBLE was very low. When the result accuracy of the method is poor, this method is not representative. If the results of different methods are very different, these methods are not comparable at all. Therefore, the effectiveness of the comparative experiment is questionable, and all the comparative experimental analysis should be carried out again.

Lines239-240  What is the basis for “In the first (resp. second) experiment we consider the Cereals/Oilseeds resp. Forest) land cover as positive class while the rest is considered as negative class.”

Author Response

The response to Reviewer 1 are in the attached file

Reviewer 2 Report

1) The methodology part is very weak. There is no explanation about the architecture and training algorithm of RNN. A diagram and mathematical equations about recurrent neural networks must be included.

2) The authors also must state why RNN is preferred over many other good deep learning based classification approaches. Is it because of time-series data? If yes, why not LSTM??

3) The confusion matrices of 4 methods are shown here. It means the authors have implemented 4 methods. But, the methodology part is without the other three methods.

4) Why haven't you estimated sensitivity and specificity when the objective is classification?

5) Accuracy is good but not good enough when you have an imbalanced dataset. The number of samples of the positive class and negative class must be given.

6) If you have the positive class, then the rest can be treated as negative class. Do you really need auto-encoders to estimate the negative class? How much improvement do you get when you use auto-encoders in comparison to not using auto-encoders?

7) How does the performance measures change when the number of samples change? More clarity is needed on that. When the same class samples are included, I feel more difference must not be there in performance measures

8) Have you compared your method with state-of-art classification methods? It is mandatory.

9) section 4.1 is ideal for literature survey section

Author Response

The responses to Reviewer 2 are in the attached file

Reviewer 3 Report

The article introduces a novel methodology of Positive Unlabelled Learning for Satellite Images Time Series analysis. It is very clear, very well explained. With very well performed and contrasted experimental results.

Then, some minor issues to review are indicated:

I think it would be useful to explain better how the reconstruction error used to differentiate the NRs is obtained. What threshold value is used to fix the red vertical line in figure 1?  I guess it has to do with the statement on line 141 that says "average reconstruction error computed on the set of unlabelled samples U and," But some further explanation would be interesting.

I think the statement on line 135 about setting the threshold to 1 should be explained in more detail.

In the caption of figure 3, I think it would be convenient to specify the contents of subfigures a), b) and c), rather than under each one. The same goes for figures 6, 7, 8 and 9.

Very good work, with magnificent results compared to the state of the art. I congratulate the authors for a job well done.

Author Response

The responses to Reviewer 3 are in the attached file

Round 2

Reviewer 1 Report

When the result accuracy of the method is poor, this method is not representative. If the results of different methods are very different, these methods are not comparable at all. Therefore, the effectiveness of the comparative experiment is questionable, and all the comparative experimental analysis should be carried out again. Other classical methods can be used for comparative analysis.

Author Response

We thank the Reviewer for this remark and we hope we will able to clarify this point. As highlighted during the first round of review, our work has the objective to cope with the Positive and Unlabelled Learning (PUL) setting in the particular case of satellite image time series classification. To the best of our knowledge, only a limited number of approaches were proposed to tackle such a setting in the field of remote sensing [1–3] and, unfortunately, none of them was especially conceived to manage satellite image time series data. For this reason, in our experimental evaluation, we have considered standard (RFPUL and OCSVM) [1] as well as recent [2] frameworks, that can fit our setting, as competing methods to evaluate the performances of our proposal. The experimental evaluations highlight that previously PUL frameworks, introduced in the remote sensing fields, were not able to provide satisfactory results in the context of satellite image time series. This fact clearly supports the foundation of our research study and it underlines the necessity to conceive tailored strategies to cope with the complexity of satellite image time series data.

[1] Li, W.; Guo, Q.; Elkan, C. A Positive and Unlabeled Learning Algorithm for One-Class Classification of Remote-Sensing Data. IEEE Trans. Geosci. Remote. Sens. 2011, 49, 717–725.

[2]  Ran, L.; Li, W.; Liu, X.; Lu, X.; Li, T.; Guo, Q. An Ensemble of Classifiers Based on Positive and Unlabeled Data in One-Class Remote Sensing Classification. IEEE J. Sel. Top. Appl. Earth Obs. Remote. Sens. 2018, 11, 572–584.

[3] Bekker, J.; Davis, J. Learning from positive and unlabeled data: a survey. Machine Learning 2020, 109, 719–760.

Reviewer 2 Report

The paper is improved. It can be accepted 

Author Response

We thank again the reviewer for his/her useful remarks that allow us to ameliorate the quality of our manuscript.
